

# Modernising tactile acuity assessment; clinimetrics of semi-automated tests and effects of age, sex and anthropometry on performance

Nick A. Olthof[1,2], Michel W. Coppieters[2,3], G Lorimer Moseley[4], Michele Sterling[5,6], Dylan J. Chippindall[2] and Daniel S. Harvie[1,2,4]

[1] School of Health Sciences and Social Work, Griffith University, Brisbane and Gold Coast, QLD, Australia
[2] Menzies Health Institute Queensland, Griffith University, Brisbane and Gold Coast, QLD, Australia
[3] Amsterdam Movement Sciences, Faculty of Behavioural and Movement Sciences, Vrije Universiteit Amsterdam, Amsterdam, The Netherlands
[4] IIMPACT in Health, University of South Australia, Kaurna Country, Adelaide, SA, Australia
[5] RECOVER Injury Research Centre, The University of Queensland, Brisbane, QLD, Australia
[6] NHMRC Centre of Research Excellence in Road Traffic Injury Recovery, The University of Queensland, Brisbane, QLD, Australia

Corresponding author
Daniel S. Harvie,
daniel.harvie@unisa.edu.au

## ABSTRACT

**Background.** Reduced tactile acuity has been observed in several chronic pain conditions and has been proposed as a clinical indicator of somatosensory impairments related to the condition. As some interventions targeting these impairments have resulted in pain reduction, assessing tactile acuity may have significant clinical potential. While two-point discrimination threshold (TPDT) is a popular method of assessing tactile acuity, large measurement error has been observed (impeding responsiveness) and its validity has been questioned. The recently developed semi-automated 'imprint Tactile Acuity Device' (iTAD) may improve tactile acuity assessment, but clinimetric properties of its scores (accuracy score, response time and rate correct score) need further examination.

**Aims.** Experiment 1: To determine inter-rater reliability and measurement error of TPDT and iTAD assessments. Experiment 2: To determine internal consistencies and floor or ceiling effects of iTAD scores, and investigate effects of age, sex, and anthropometry on performance.

**Methods.** Experiment 1: To assess inter-rater reliability ($ICC_{(2,1)}$) and measurement error (coefficient of variation (CoV)), three assessors each performed TPDT and iTAD assessments at the neck in forty healthy participants. Experiment 2: To assess internal consistency ($ICC_{(2,k)}$) and floor or ceiling effects (skewness z-scores), one hundred healthy participants performed the iTAD's localisation and orientation tests. Balanced for sex, participants were equally divided over five age brackets (18–30, 31–40, 41–50, 51–60 and 61–70). Age, sex, body mass index (BMI) and neck surface area were assessed to examine their direct (using multiple linear regression analysis) and indirect (using sequential mediation analysis) relationship with iTAD scores.

**Results.** Mean $ICC_{(2,1)}$ was moderate for TPDT (0.70) and moderate-to-good for the various iTAD scores (0.65–0.86). The CoV was 25.3% for TPDT and ranged from 6.1% to 16.5% for iTAD scores. Internal consistency was high for both iTAD accuracy scores ($ICC_{(2,6)} = 0.84$; $ICC_{(2,4)} = 0.86$). No overt floor or ceiling effects were detected (all
skewness z-scores < 3.29). Accuracy scores were only directly related to age (decreasing with increasing age) and sex (higher for men).

**Discussion**. Although reliability was similar, iTAD scores demonstrated less measurement error than TPDT indicating a potential for better responsiveness to treatment effects. Further, unlike previously reported for TPDT, iTAD scores appeared independent of anthropometry, which simplifies interpretation. Additionally, the iTAD assesses multiple aspects of tactile processing which may provide a more comprehensive evaluation of tactile acuity. Taken together, the iTAD shows promise in measuring tactile acuity, but patient studies are needed to verify clinical relevance.

## BACKGROUND

Measures of tactile acuity have been utilised to identify somatosensory impairments in a variety of painful conditions, such as musculoskeletal disorders (*Debenham et al., 2016*; *Fujimoto & Kon, 2016*; *Mena-Del Horno et al., 2020*), chronic pain (*Catley et al., 2014*; *Harvie, Edmond-Hank & Smith, 2018*), amputation (*Vega-Bermudez & Johnson, 2002*; *Guemann et al., 2019*), neuropathy (*Fonseca et al., 2018*), stroke (*Rinderknecht et al., 2019*), arthritis (*Stanton et al., 2013*), complex regional pain syndrome (*Lewis & Schweinhardt, 2012*) and spinal cord injury (*Zeilig et al., 2012*). Patients with persistent pain typically demonstrate poorer tactile acuity than healthy controls (*Catley et al., 2014*), which is thought to reflect changes in somatosensory processing related to the condition. Since some interventions targeting these impairments have shown promise (*Pleger et al., 2005*; *Moseley, Zalucki & Wiech, 2008*; *Schmid et al., 2017*; *Wakolbinger et al., 2018*; *Wand et al., 2013*), assessing tactile acuity may have significant clinical potential.

Tactile acuity refers to the accuracy and clearness of touch perception (*Walter, 2008*). Several dimensions of tactile acuity have been identified, such as two-point (or gap) discrimination, point (mis)localisation, length discrimination, orientation discrimination and shape/texture recognition (*Marsh, 1990*; *Bell-Krotoski, Weinstein & Weinstein, 1993*; *Stevens & Patterson, 1995*). As cutaneous mechanoreceptors respond to a variety of tactile stimuli (*Abraira & Ginty, 2013*), tactile acuity measures apply either dynamic deformation (movement), indentation (pressure) or vibration to the skin (*Demain et al., 2013*).

The two-point discrimination threshold (TPDT), *i.e.,* the minimum distance between two tactile stimuli that can be perceived as spatially distinct, is the most frequently used and studied measure of tactile acuity (*Catley et al., 2014*; *Fonseca et al., 2018*; *Ehrenbrusthoff et al., 2018*). Despite its popularity, its validity has been questioned due to involvement of non-spatial cues (*Boldt et al., 2014*; *Craig & Johnson, 2000*; *Lundborg & Rosen, 2004*; *Tong, Mao & Goldreich, 2013*), and observations of abnormal scores (*Craig & Johnson, 2000*; *Lundborg & Rosen, 2004*), hindering its interpretation. Moreover, TPDT scores appear affected by age (*Marsh, 1990*; *Stevens & Patterson, 1995*; *Kalisch et al., 2009*; *Kalisch et al.,*

*2012*), sex (*Kalisch et al., 2012*; *Falling & Mani, 2016b*), and anthropometry (*e.g.*, body mass index (BMI), waist-hip ratio, surface area of the tested body part) (*Falling & Mani, 2016b*; *Peters, Hackeman & Goldreich, 2009*), which further complicates interpretation of results. Although moderate to good intra-rater and inter-rater reliability has been reported, large measurement error has been observed (*Catley et al., 2013*), making it difficult to detect treatment effects. Correspondingly, TPDT was the least responsive sensibility test following median nerve injury and repair (*Fonseca et al., 2018*; *Jerosch-Herold, 2003*), as well as the least responsive to improvements in hand function after surgery (*Fujimoto & Kon, 2016*).

A variety of tests have recently been developed to improve tactile acuity assessment, such as the two-point orientation test (*Tong, Mao & Goldreich, 2013*), point-to-point test (*Adamczyk et al., 2016*), tactile acuity charts (*Bruns et al., 2014*), grating orientation task (*Van Boven et al., 2000*), and two-point estimation task (*Adamczyk et al., 2019b*; *Zimney et al., 2020*). Additionally, technological developments instigated semi-automated tests which may be less affected by inter-rater variability (*Guemann et al., 2019*; *Rinderknecht et al., 2019*; *Hoffmann et al., 2018*). An added benefit of these semi-automated tools is the potential for independent sensory training. This may be of clinical relevance, given that treatment success in manual sensory discrimination interventions could be limited by the need for caregiver involvement during at home training (*Ryan et al., 2014*). However, despite neck pain being ranked among the top five leading causes of years lived with disability globally (*Global Burden of Disease Study 2013 Collaborators, 2015*), only a limited number of these novel procedures have been applied to the neck (*Zimney et al., 2020*; *Harvie et al., 2017*; *Morrow & Ziat, 2018*; *Adamczyk et al., 2019a*). As such, the body of knowledge about tactile acuity in neck pain appears limited compared to other painful conditions (*Luedtke & Adamczyk, 2017*). The development of improved tactile acuity assessment at the neck therefore has the potential to elucidate mechanisms of neck pain, inform development of new treatment strategies, and guide clinical decision making.

One recently developed tool designed to assess tactile acuity at the neck is the 'Imprint Tactile Acuity Device' (iTAD) (*Olthof et al., 2021*). The iTAD is a semi-automated device that uses single and successive vibrotactile stimuli to quantify absolute and relative tactile localisation performance (for details see elsewhere (*Olthof et al., 2021*)). As stimulus administration is automated and does not require synchronicity between locations, the iTAD may overcome some complications associated with TPDT. Despite some initial design issues, the iTAD prototype has shown comparable intra-rater reliability to existing tests (*Olthof et al., 2021*), suggesting prospective utility. In this manuscript, we report two experiments that investigate the clinimetric properties of an updated version of the iTAD. Experiment 1 aimed to determine inter-rater reliability and measurement error of both the iTAD and TPDT assessments, which is currently the most reliable tactile acuity assessment at the neck (*Harvie et al., 2017*). Experiment 2 aimed to quantify internal consistency and identify floor or ceiling effects of the iTAD scores, and determine their relationship with age, sex, BMI and neck surface area.

## METHODS

Both experiments were approved by Griffith University Human Research Ethics committee (#2016/168), and all participants gave written informed consent prior to participating.

### Experiment 1: Inter-rater reliability and measurement error
#### Design and procedure

A test-retest study investigated the inter-rater reliability and measurement error of the iTAD and TPDT. Three assessors each performed both assessments in a private, quiet, room. The order of assessments was randomised between participants, but consistent between assessors for each participant. The order of assessors was randomised between participants. Participants received a 10-minute break between assessors and were blinded to their scores. Assessors were blinded to all iTAD scores and TPDT scores of the other assessors. The anatomical location of both assessments was standardised but independently determined by each assessor.

#### Participants

Using a convenience sample, individuals without current pain and neurological symptoms as well as without a history of persistent (*i.e.,* >3 months) pain and neurological symptoms in the past five years were recruited from the general public.

#### Assessors

Three final year physiotherapy Masters students each performed iTAD and TPDT assessments after receiving approximately one hour of training for each procedure. Test instructions and performance were standardised using a testing protocol. Training included approximately 20–30 min of instructions on the testing protocol and about 30–40 min of practicing the protocol. Each assessor practised each protocol five times and underwent each assessment at least once.

#### TPDT assessment

A digital caliper (Renegade industrial, carbon fibre, RCFVC150) was used to establish TPDT, utilizing the two arms and pressure of its own weight as tactile stimuli. Contact area of the tip of each arm was approximately 0.25 mm × 0.5 mm and stimulus duration <1 s. During TPDT assessment, participants were seated with their forehead resting on a table in front of them. On the dominant side, TPDT was measured in a cranio-caudal direction with the caudal arm of the caliper stationary at 15 mm lateral to the spinous process of C7.

Similar to a previously published procedure (*Luedtke et al., 2018*), a two-alternative forced-choice (one or two points) staircase method was used, alternating two ascending and two descending runs (see Fig. 1). The caliper distance started at 15 mm, increasing with five mm steps. Steps were reduced to two mm for the subsequent runs. To avoid guessing, three consecutive reports of either one or two points indicated a reversal. A 10 mm step was added in the direction of the completed run, before running the reversed direction. The TPDT was calculated by averaging the scores of the four reversals, expressed in millimetres, with larger distances indicating poorer tactile acuity.

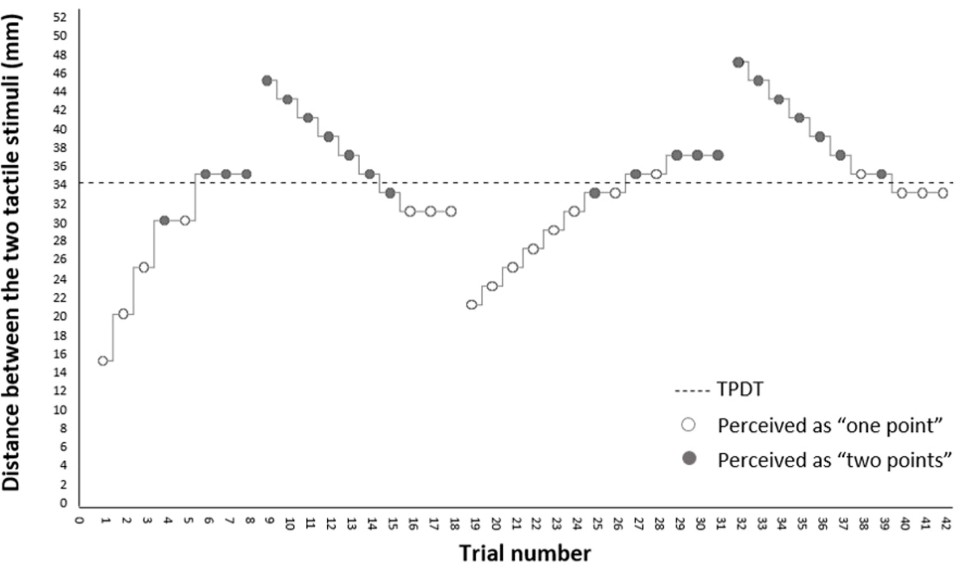

**Figure 1** **TPDT assessment procedure.** Example of the assessment of a hypothetical two-point discrimination threshold (TPDT). Assessment is based on a forced-choice response (one or two points), alternating four runs with either increasing or decreasing caliper distances. Steps taken are either in five mm. (first run) or two mm. (other runs). Three consecutive reports of either one or two points indicates a reversal. Mean of the four reversals is calculated for the TPDT score.

### iTAD assessment

*iTAD prototype.* The iTAD consists of a wearable neoprene collar containing twelve vibrotactile stimulators (~200 Hz with 0.75 g vibration amplitude), arranged in three rows of four (see Fig. 2). A wirelessly connected tablet operates the stimulators and records user's responses. After fitting and familiarisation, two tactile acuity tests were performed: the localisation test, which measures the ability to localise the vibrations (one second stimulus duration), and the orientation test, which measures the ability to determine the orientation of two successive adjacent vibrations relative to each other (0.7 s stimulus duration each). Accuracy scores (*i.e.,* percentage correct) for each test, and the overall score (*i.e.,* mean of both test), were calculated with higher scores indicating better tactile acuity. For a full description of the iTAD prototype and its assessment procedures, see elsewhere (*Olthof et al., 2021*).

*Changes to the prototype.* After development of the prototype, the internode distance was reduced to 32.5 mm (centre-to-centre) between all rows and columns. Additionally, a layer of foam was placed in the collar to aid fitting consistency. For the localisation test, the number of trials was increased from 48 to 72, delivered in six series of twelve. For the orientation test, the number of trials was increased from 48 to 64, delivered in four series of sixteen. Furthermore, trials were block randomised within each series, alternating between sides of the neck.

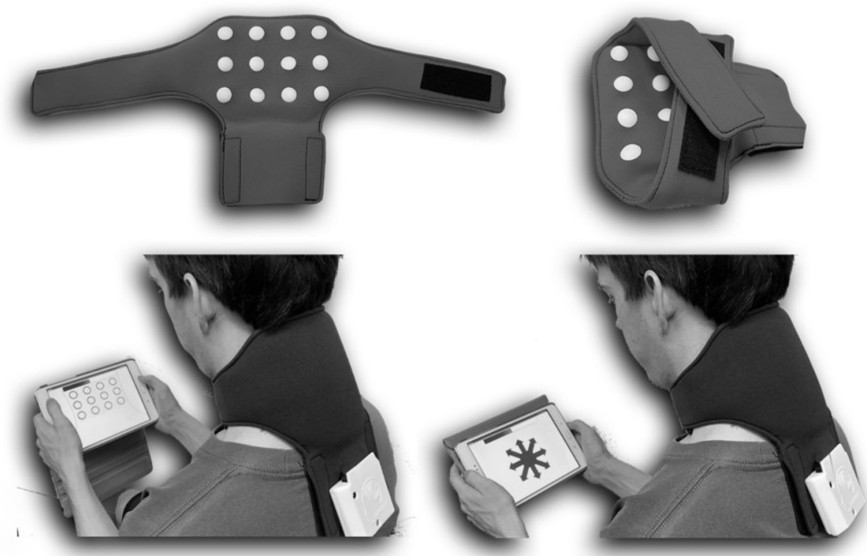

**Figure 2** **The imprint Tactile Acuity Device (iTAD), containing twelve build-in vibrotactile stimulators (top), and wirelessly connected tablet.** The iTAD performs two tactile acuity tests: (1) the localisation test (bottom left) where the perceived location of the tactile stimulus is selected and (2) the orientation test (bottom right) where the perceived location of a second tactile stimulus, relative to a first, is selected. For both tests, as well as the overall score (mean of both tests), the accuracy score (*i.e.,* percentage correct), the average response time and the rate correct score (number of correct responses per minute of response activity) is calculated.

*Additional scores.* In order to better quantify tactile acuity, two new scores were added. For each test, and the overall score, (average) response time was recorded in milliseconds with lower scores indicating faster responses. Additionally, a rate correct score was calculated $(= \sum(\text{correct responses})/\sum(\text{response times in minutes}))$, quantifying the number of correct responses per minute of response activity (*Vandierendonck, 2017*) with higher scores indicating better tactile acuity. As the rate correct score integrates response time and accuracy, it accounts for individual speed-accuracy trade-off strategies and has been suggested to provide a better estimate of perceptual performance (*Vandierendonck, 2017*).

### Statistical analyses

Inter-rater reliability was assessed by calculating the intraclass correlation coefficient, model 2,1 ($\text{ICC}_{(2,1)}$) (*i.e.,* two-way random, absolute agreement, single measures). Values were interpreted as poor (<0.5), moderate (0.5–0.74), good (0.75–0.89) or excellent (≥0.9) (*Portney & Watkins, 2009*).

The standard error of measurement (SEM) was calculated using variance components from the ANOVA table ($\sqrt{(\sigma^2_{observer} + \sigma^2_{residual})}$) to assess measurement error (*De Vet et al., 2006*). As TPDT and iTAD use different metric units, the SEM was additionally converted to the coefficient of variation (CoV) (*i.e.,* SEM as a percentage of mean score) allowing direct comparison of measurement error (*Hopkins, 2000*). Since both <10% and

<20% have been suggested as a good CoV (*Atkinson & Nevill, 1998*; *Quan & Shih, 1996*), results were categorised into: <10%, 10–20% and >20%.

To assist clinical interpretation of the SEM, the smallest detectable change with a 95% confidence interval (SDC_95) was calculated (1.96*$\sqrt{2}$*SEM) (*De Vet et al., 2006*). Additionally, SDC values with smaller confidence intervals (SDC_80, SDC_85 and SDC_90) were computed (replacing 1.96 with respective z-values).

### Sample size calculation

To detect a hypothesized $ICC_{(2,1)}$ of 0.7 with three repeated measurements (recommended to optimise sample size (*Walter, Eliasziw & Donner, 1998*)), while $\alpha = 0.05$, $\beta = 0.2$ and $ICC_{(2,1)} = 0.5$ for the null hypothesis, a minimum of 40 participants is required (*Walter, Eliasziw & Donner, 1998*). Therefore, we aimed to recruit 40 participants.

## Experiment 2: Internal consistency, floor and ceiling effects, and relationship with age, sex and anthropometry
### Design and procedure

Using a cross-sectional design, the internal consistencies of the localisation and orientation accuracy score were investigated. Additionally, floor and ceiling effects of all iTAD scores were assessed, as well as their relationship with age, sex, BMI and neck surface area. In one session, participants performed both iTAD tests after age, sex, and anthropometric measures (see 'Anthropometric measurements') were recorded. Measures were taken by a single assessor in a private, quiet room. The assessor had several hours of prior experience performing iTAD assessments.

### Participants

Recruitment and selection criteria were identical to experiment one. However, participants represent a different cohort without overlap.

### iTAD tests

For procedure of the iTAD tests, see 'iTAD assessment'.

### Anthropometric measurements

After weight (kg) and height (m) were recorded, BMI was calculated (weight/height$^2$). Using a tape measure, the distance from the caudal aspect of the external occipital protuberance to the spinous process of C7 was measured to quantify neck length (cm). Neck circumference (cm) was measured at half-way of the neck length measurement placing the tape measure horizontally around the neck. Using these measurements, the posterior neck surface area was estimated (neck length*(neck circumference/2)).

### Statistical analysis

Internal consistency was assessed by calculating the inter-relatedness of accuracy scores between series within each test, using ICC model 2,k (*i.e.,* two-way random, absolute agreement, average measures). The $ICC_{(2,k)}$ was chosen over the Cronbach's alpha, the equivalent of the $ICC_{(3,k)}$ (*i.e.,* two-way random, consistency, average measures), to include absolute differences between series. Although various cut-offs are proposed, most recommend 0.7−0.9 for high internal consistency (*Taber, 2018*).

For the accuracy scores, floor and ceiling effects were considered present if >15% of the participant scored within either the highest or lowest 20% of the scale (*Terwee et al., 2007*). However, such assessment would not be adequate for response times or rate correct scores, as their scales have no limit on one end and scores are (near) impossible at the other. Therefore, floor and ceiling effects for all iTAD scores were assessed by calculating z-scores for the skewness of their distribution (*i.e.*, the skew value divided by its standard error) (*Kim, 2013*; *Ho & Yu, 2015*). Floor and ceiling effects may be present with a z-score >±1.96 in small (n<50) or >±3.29 in medium (50<n<300) sized samples (*Kim, 2013*; *Ho & Yu, 2015*).

In order to study the direct multivariate relationships of age, sex, BMI and neck surface area with the localisation and orientation accuracy score, multiple linear regressions (enter models) were performed. Furthermore, to estimate the potential indirect effects of age and sex through BMI and/or neck surface area, sequential mediation analyses were performed using the SPSS extension PROCESS (model 6; 5000 bootstrapped samples) as a secondary analysis. Mediation analyses for all other iTAD scores were performed as exploratory analyses. Regression models are expressed in (adjusted) explained variance ($R^2_{adjusted}$). For all relationships, both the mean unstandardized regression coefficient (b) and the semi-partial correlation (sr) are provided. Indirect effects are expressed in percentage mediation ($P_m$).

### Sample size

To examine internal consistency, a minimum of 100 participants is recommended (*Terwee et al., 2007*). To explore potential floor and ceiling effects, a minimum of 50 participants is recommended (*Terwee et al., 2007*). For the multiple linear regressions, 100 participants were needed to find a medium sized ($f^2 = 0.15$) prediction model using four predictors with a Bonferroni corrected *p*-value of 0.025 and 80% power. Taken together, the sample size was set for 100 participants, with 10 participants of both sexes in each of five age brackets (18–30, 31–40, 41–50, 51–60 and 61–70).

## RESULTS

### Experiment 1: Inter-rater reliability and measurement error

Forty individuals (25 male) participated, with a mean (SD) age of 24.1 (4.5) years. One participant was left hand dominant and the others right hand dominant. Mean scores, inter-rater reliabilities and measurement errors are displayed in Table 1. Inter-rater reliability was good for iTAD orientation accuracy score, overall accuracy score, and all response times. All other scores displayed moderate inter-rater reliability. The CoV was <10% for all iTAD response times and >20% for TPDT. All other scores had a CoV of 10–20%.

For all scores, the SDC is presented with varying confidence intervals (80% to 95%) in Table 2. Each can be used to assess the chance that an observed change in score, when larger in either direction, could reflect measurement error: (100%–confidence interval)/2 (*i.e.*, <10%, <7.5%, <5% and <2.5% respectively). For example, a change of +16.2% in

**Table 1** Mean, inter-rater reliability and measurement error for the iTAD and two-point discrimination threshold scores.

| | Score | Score | Inter-rater reliability | Measurement error | |
| --- | --- | --- | --- | --- | --- |
| | Metric | Mean (SD) | $ICC_{(2,1)}$(95% CI) | SEM | CoV |
| **iTAD Localisation test** | AS | 61.0% (12.7) | 0.65 (0.49–0.78) | 8.7% | 14.3% |
| | RT | 1238.5 ms (241.9) | 0.82 (0.71–0.90) | 109.7 ms | 8.9% |
| | RCS | 30.5 c/min (7.5) | 0.65 (0.49–0.78) | 5.0 c/min | 16.5% |
| **iTAD Orientation test** | AS | 46.2% (10.9) | 0.76 (0.63–0.85) | 5.9% | 12.7% |
| | RT | 1995.4 ms (281.7) | 0.80 (0.69–0.88) | 134.0 ms | 6.7% |
| | RCS | 14.0 c/min (3.3) | 0.74 (0.61–0.84) | 1.9 c/min | 13.2% |
| **iTAD Overall score** | AS | 53.6% (10.4) | 0.75 (0.61–0.85) | 5.8% | 10.8% |
| | RT | 1617.0 ms (246.4) | 0.86 (0.76–0.92) | 97.8 ms | 6.1% |
| | RCS | 20.2 c/min (4.0) | 0.72 (0.58–0.83) | 2.3 c/min | 11.6% |
| **TPDT** | Distance | 47.7 mm (19.5) | 0.70 (0.55–0.81) | 12.1 mm | 25.3% |

Notes.

iTAD, imprint Tactile Acuity Device; TPDT, two-point discrimination threshold; AS, accuracy score; RT, average response time; RCS, rate correct score; ms, milliseconds; c/min, correct responses per minute; ICC, intraclass correlation coefficient; SEM, standard error of measurement; CoV, coefficient of variation.

**Table 2** Smallest detectable changes with 80%, 85%, 90% and 95% confidence intervals for the iTAD and two-point discrimination threshold scores.

| | Score | SDC_80 | SDC_85 | SDC_90 | SDC_95 |
| --- | --- | --- | --- | --- | --- |
| **iTAD Localisation test** | AS (%) | 15.7 | 17.7 | 20.2 | 24.1 |
| | RT (ms) | 198.5 | 223.2 | 255.2 | 304.0 |
| | RCS (c/min) | 9.1 | 10.2 | 11.7 | 13.9 |
| **iTAD Orientation test** | AS (%) | 10.7 | 12.0 | 13.7 | 16.3 |
| | RT (ms) | 242.5 | 272.7 | 311.7 | 371.4 |
| | RCS (c/min) | 3.4 | 3.8 | 4.3 | 5.1 |
| **iTAD Overall score** | AS (%) | 10.5 | 11.8 | 13.5 | 16.0 |
| | RT (ms) | 177.1 | 199.1 | 227.6 | 271.2 |
| | RCS (c/min) | 4.2 | 4.8 | 5.4 | 6.5 |
| **TPDT** | Distance (mm) | 21.9 | 24.6 | 28.1 | 33.5 |

Notes.

iTAD, imprint Tactile Acuity Device; TPDT, two-point discrimination threshold; AS, accuracy score; RT, average response time; RCS, rate correct score; ms, milliseconds; c/min, correct responses per minute; SDC, smallest detectable change.

localisation accuracy score would have a 7.5–10% chance to be a result of measurement error, whereas a change of +21.3% a 2.5–5% chance.

## Experiment 2: Internal consistency, floor and ceiling effects, and relationship with age, sex and anthropometry

One hundred individuals participated, with ten of both sexes per predetermined age bracket. Ten participants were left hand dominant, 88 right hand dominant and two were ambidextrous. Mean (SD) BMI was 26.4 (4.6) and mean (SD) neck surface area was 261.0 cm$^2$ (48.1). Mean (SD) duration was 03:06 (00:25) minutes for the localisation test and 03:38 (00:28) for the orientation test.

### Internal consistency

The $ICC_{(2,6)}$ (95% CI) for the localisation accuracy score was 0.84 (0.79−0.89) and the $ICC_{(2,4)}$ (95% CI) for the orientation accuracy score was 0.86 (0.81−0.90).

### Floor and ceiling effects

None of the accuracy scores had >5% of participants scoring in either the highest or lowest 20%. Only localisation response time had a skewness z-score $>\pm1.96$ (z $=+2.73$), which was still $<\pm3.29$.

### Relationship with age, sex and anthropometry

Multiple linear regression analyses showed significant prediction models for both the localisation accuracy score (F(4,95) $=4.92$, $p < 0.01$, $R^2_{adjusted} = 0.14$) and orientation accuracy score (F(4,95) $=5.82$, $p < 0.01$, $R^2_{adjusted} = 0.16$).

For the localisation accuracy score, both age (b $=-0.20$, sr $=-0.20$, $p = 0.03$) and sex (b $= 8.04$, sr $=0.25$, $p = 0.01$) had a significant contribution to the model, whereas BMI ($p = 0.11$) and neck surface area ($p = 0.89$) did not. This indicates that, on average, men scored 8.04% higher than women, and that scores decreased by 0.20% for each year of age.

For the orientation accuracy score, both age (b $=-0.39$, sr $=-0.36$, $p = 0.00$) and sex (b $= 6.69$, sr $=0.19$, $p = 0.04$) contributed significantly to the model, whereas BMI ($p = 0.71$) and neck surface area ($p = 0.94$) did not. This indicates that, on average, men scored 6.69% higher than women, and that scores decreased by 0.39% for each year of age.

The mediation analyses indicated several significant relationships between demographic and anthropometric variables (see Fig. 3). However, for the localisation accuracy score, the total indirect effects of age ($P_m = 0.16$, $p > 0.05$), and sex ($P_m = 0.06$, $p > 0.05$), through BMI and neck surface area were non-significant. Similarly, the total indirect effects of age ($P_m = 0.02$, $p > 0.05$), and sex ($P_m = 0.04$, $p > 0.05$) were also non-significant for the orientation accuracy score. Additionally, all individual indirect effects were non-significant for both tests. This indicates that BMI and/or neck surface area did not significantly mediate the effects of age or sex for either accuracy score. Scatterplots of localisation and orientation accuracy scores as a function of age and sex are displayed in Fig. 4.

Exploratory mediation analysis of the other iTAD scores showed similar patterns, other than sex not significantly predicting the localisation or orientation response time and rate correct score. Additionally, rate correct scores were more strongly predicted by age. Figures of all mediation analyses can be found in Supplemental files (Figs. S1–S3). Scatterplots of all iTAD scores as a function of age and sex can be also be found in Supplement files (Figs. S4, S5).

## DISCUSSION

### Inter-rater reliability

Inter-rater reliability was moderate for TPDT and moderate to good for the iTAD scores. When directly compared, $ICC_{(2.1)}$ values were somewhat higher for iTAD's response times, but similar between TPDT and other iTAD scores. The $ICC_{(2.1)}$ values for TPDT appear comparable to previous research, although results vary (*Catley et al., 2013*; *Harvie et al.,*

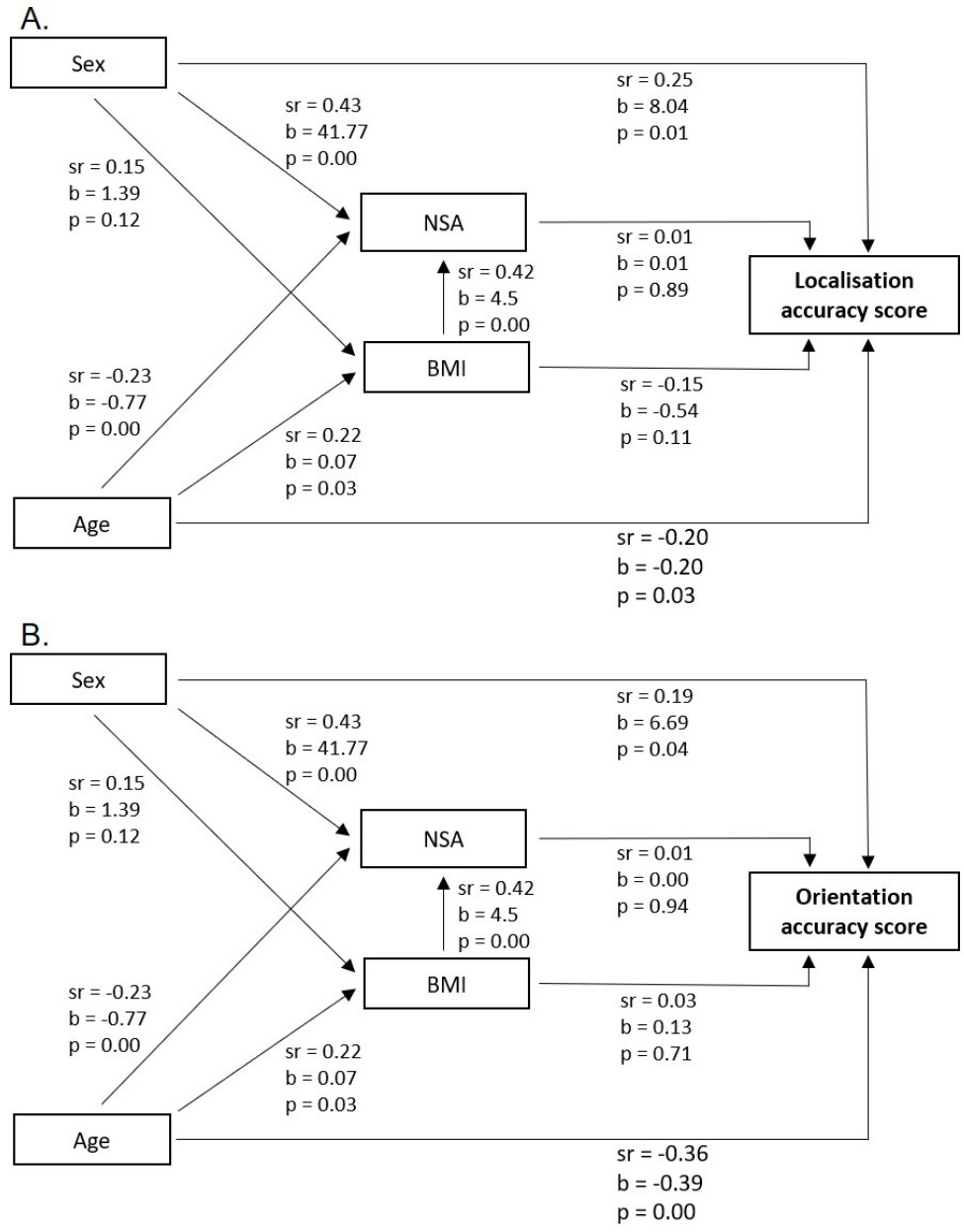

**Figure 3 Results sequential mediation analyses.** Relationships between demographics (sex and age), anthropometrics (body mass index (BMI) and neck surface area (NSA)) and iTAD accuracy scores for the localisation test (A) and orientation test (B). Relationships are expressed in semi-partial correlations (sr) and unstandardized regression coefficients (b), including their level of significance (p). Coding for sex: female =0 and male =1.

*2017*; *Luedtke et al., 2018*; *Adamczyk, Luedtke & Szikszay, 2018*; *Cashin, 2017*). Compared to the prototype, reliability appears improved with the new generation iTAD despite previous values representing intra-rater reliability (*Olthof et al., 2021*).

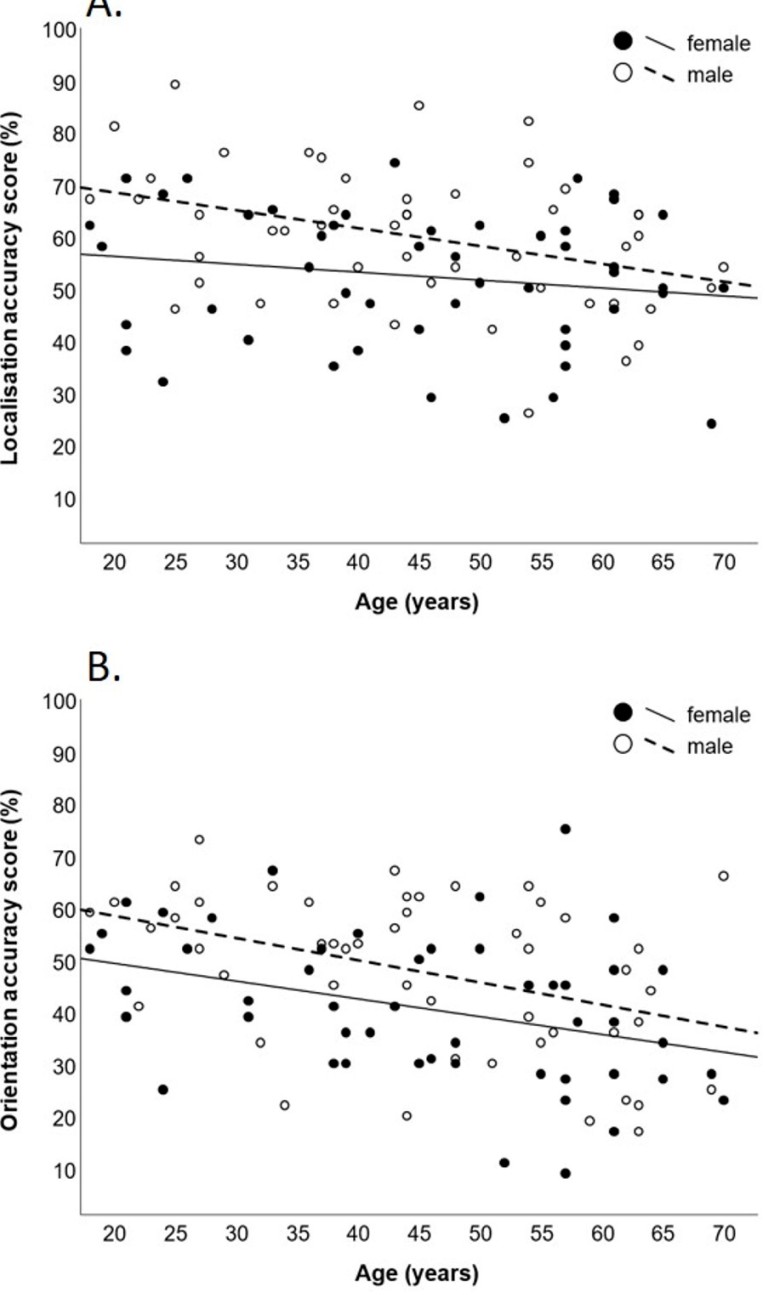

**Figure 4 Scatter plots of localisation and orientation accuracy scores.** Scatter plots of iTAD accuracy scores as a function of age and sex. Scores are displayed for the localisation (A) and orientation (B) test. Lines represent the least squares regressions.

## Measurement error

Results indicate a larger CoV for TPDT than all iTAD scores. For TPDT assessment, differences in speed, timing and intensity of stimulus delivery affect results (*Boldt et al., 2014*; *Lundborg & Rosen, 2004*; *Yokota et al., 2020*). Therefore, variability in these

parameters between raters, trials, and both arms of the caliper, increases measurement error. An inherent problem of manual TPDT assessment is the inability to control, or assess, these variables in a clinical setting. The iTAD scores may be less prone to these sources of error variance, as stimulus administration is automated and does not require synchronicity between locations. Clinically, this implies less difficulty detecting change with iTAD assessments, potentially resulting in better responsiveness to treatment effects (*Terwee et al., 2007*). Notably, CoV for TPDT seems somewhat larger than previously reported (17.4–20.6% (*Luedtke et al., 2018*); 19.1% (*Catley et al., 2013*)). This may be due to the testing procedure (*e.g.*, orientation of caliper, number of reversals), for which no standard is available (*Adamczyk, Luedtke & Szikszay, 2018*; *Cashin, 2017*). Yet, mean (SD) TPDT scores appeared similar to several other reports (mean (SD) range: 45.9 (18.4) to 62.6 (22.9)) (*Catley et al., 2013*; *Zimney et al., 2020*; *Adamczyk et al., 2019a*; *Cheever et al., 2017*), although somewhat higher than others (mean (SD) range: 21.7 (6.2) to 35.2 (9.6)) (*Harvie et al., 2017*; *Luedtke et al., 2018*; *Elsig et al., 2014*). Further, measurement error may depend on various factors related to both participants and assessors included (*Catley et al., 2013*). However, these were constant between the two assessments in this experiment, allowing for a more direct comparison.

Several SDC values were also presented. Although conventional, the SDC_95 may provide high specificity (few false positives) but low sensitivity (many false negatives) in detecting change due to its large confidence interval (*Portney & Watkins, 2009*). As both false conclusions can negatively impact clinical decision making, presenting a range of SDC values may support more precise interpretation of observed changes in relation to measurement error.

### Internal consistency

Despite including absolute differences between series, internal consistency was high for both the localisation and orientation accuracy scores and higher than for the iTAD prototype (*Olthof et al., 2021*). Internal consistency is frequently applied to questionnaires but underutilised in experimental tasks, mostly because scores cannot be split into multiple representative parts (*Green et al., 2016*; *Matheson, 2019*). However, internal consistency has previously been established in measures such as electrocardiography (*Van Lien et al., 2015*), electroencephalography (*Towers & Allen, 2009*), joint position sense (*Domingo & Lam, 2014*) and motion analysis (*Platz et al., 1999*). One benefit of reporting internal consistency as a measure of reliability is its comparability between studies, even if only single measurements are taken (*Green et al., 2016*).

### Floor and ceiling effects

No floor or ceiling effects, which may limit responsiveness (*Terwee et al., 2007*), were detected in iTAD scores; only a potential, yet debatable, floor effect for localisation response time. This could indicate difficulty in detecting improvements in already fast responders. However, this may not be clinically relevant, as slow responders are more likely targets for treatment.

## Relationship with age, sex, BMI and neck surface area

Results indicated that localisation and orientation accuracy scores were only directly related to age (decreasing with increasing age) and sex (lower for women). This implies that age and sex, but not BMI or neck surface area, should be considered when interpreting scores.

Regarding the effect of age, similar sized negative correlations between age and tactile acuity have previously been established using TPDT (*Kalisch et al., 2012*; *Falling & Mani, 2016a*). One frequently proposed mechanism is the decreased cortical inhibition in response to tactile stimulation associated with older age (*Kalisch et al., 2009*; *Lenz et al., 2012*; *Brodoehl et al., 2013*; *Pleger et al., 2016*). Interestingly, these age-related declines in tactile acuity can potentially be reversed with sensory training (*Pleger et al., 2016*; *Dinse et al., 2006*).

Concerning the effect of sex, previous reports seem inconsistent and vary between body regions when measured with TPDT (*Falling & Mani, 2017*). For example, better tactile acuity has been reported for women at the orofacial region (*Won et al., 2017*) and knee (*Falling & Mani, 2016b*), for men at the knee (*Stanton et al., 2013*), and no differences were found at the lower back (*Stanton et al., 2013*; *Falling & Mani, 2016a*). To the best of our knowledge, this is the first study examining sex differences at the neck, making it difficult to compare results. Additionally, sex differences could dependent on task type. In a single experiment, women made more errors in a tactile object recognition task despite demonstrating similar TPDT scores (*Kalisch et al., 2012*).

For TPDT assessment, scores seem affected by BMI (*Falling & Mani, 2016b*; *Falling & Mani, 2016a*) and body fat ratios (*Boles & Givens, 2011*). One proposed mechanism is the sensitivity of TPDT scores to skin deformation (*Yokota et al., 2020*; *Boles & Givens, 2011*), which is affected by BMI (*Smalls, Randall Wickett & Visscher, 2006*). The iTAD uses vibrotactile stimuli rather than indentation, which may explain the contrasting results.

Results also contrast previous reports indicating that tactile acuity at the fingertips relates to surface area, potentially due its relationship with mechanoreceptor density (*Peters, Hackeman & Goldreich, 2009*). One explanation is that the utilized neck surface area assessment may be a poor proxy for receptive field configuration. Different to the fingertips, necks exhibit variation in hairy (*vs.* non-hairy) skin which typically does not contain Pacinian mechanoreceptors, known to be activated by high frequency vibrations (*Abraira & Ginty, 2013*). Proportion of hairy skin may therefore moderate the relationship between neck surface area and iTAD scores, which was not investigated in this study. Alternatively, lack of a significant relationship with surface area could also indicate that iTAD scores may be less affected by peripheral receptive field configuration. Correspondingly, the ability to accurately localise tactile stimuli may be more centrally organised (*Braun et al., 2011*), and higher order cognitive functions (such as cortical body representations) seem to play a more prominent role (*Longo, Azanon & Haggard, 2010*; *Tame, Azanon & Longo, 2019*). The iTAD may therefore be especially suited for conditions with altered body representations, such as musculoskeletal disorders (*Viceconti et al., 2020*) and persistent pain (*Tsay et al., 2015*).

### Implications and future directions

Less measurement error for the iTAD could result in better responsiveness, although this needs investigation in future trials studying treatment effects. Further, the multiple measures of the iTAD may provide a more comprehensive evaluation of tactile acuity function. However, validity of the iTAD assessments has not been thoroughly established, precluding inferences about their clinical utility in addition to, or instead of, TPDT. Moreover, their clinical relevance needs further examination in patient trials. Additionally, future studies could explore to what extent iTAD scores reflect central somatosensory processing using neuroimaging techniques. Future studies may also investigate the clinimetric properties and clinical utility of other promising manual (*Bruns et al., 2014*; *Van Boven et al., 2000*; *Morrow & Ziat, 2018*; *Bleyenheuft & Thonnard, 2007*) and automated (*Goldreich et al., 2009*) procedures, including automated TPDT (*Yokota et al., 2020*; *Frahm & Gervasio, 2021*), at the neck.

## CONCLUSION

Findings suggest that the iTAD and TPDT have similar inter-rater reliability when measuring tactile acuity at the neck in healthy individuals. However, the iTAD exhibits several advantages such as ability to assess multiple aspects of tactile acuity, less measurement error and a possibility for independent sensory training. Furthermore, no evidence was found that scores were affected by anthropometry, simplifying interpretation. Additionally, internal consistency of iTAD accuracy scores was high and no overt floor or ceiling effects were detected. These results highlight the potential clinical utility of the iTAD and support continued investigation.

### Funding

Daniel S. Harvie is supported by an Early Career Research Fellowship from the National Health and Medical Research Council of Australia (ID 1142929). G Lorimer Moseley is supported by a Leadership Investigator Grant from the National Health and Medical Research Council of Australia (ID 1178444). Michele Sterling received a Fellowship from the National Health and Medical Research Council of Australia. The funders had no role in study design, data collection and analysis, decision to publish, or preparation of the manuscript.

### Grant Disclosures

The following grant information was disclosed by the authors:
The National Health and Medical Research Council of Australia (ID 1142929).
The National Health and Medical Research Council of Australia (ID 1178444).
The National Health and Medical Research Council of Australia.

## Competing Interests

G Lorimer Moseley is an Academic Editor for PeerJ. In the last 5 years, G. Lorimer Moseley has received support from: Reality Health, Pfizer Australia, Seqirus, Kaiser Permanente, Workers' Compensation Boards in Australia, Europe and North America, AIA Australia, the International Olympic Committee, Port Adelaide Football Club and Arsenal Football Club. Professional and scientific bodies have reimbursed him for travel costs related to presentation of research on pain at scientific conferences/symposia. He has received speaker fees for lectures on pain and rehabilitation. NAO, DJC and DSH are exploring opportunities to commercialise the iTAD. All other authors declare no conflict of interest.

## Author Contributions

- Nick A. Olthof conceived and designed the experiments, performed the experiments, analyzed the data, prepared figures and/or tables, authored or reviewed drafts of the paper, and approved the final draft.
- Michel W. Coppieters conceived and designed the experiments, analyzed the data, authored or reviewed drafts of the paper, and approved the final draft.
- G Lorimer Moseley and Michele Sterling conceived and designed the experiments, authored or reviewed drafts of the paper, and approved the final draft.
- Dylan J. Chippindall conceived and designed the experiments, performed the experiments, prepared figures and/or tables, authored or reviewed drafts of the paper, and approved the final draft.
- Daniel S. Harvie conceived and designed the experiments, analyzed the data, prepared figures and/or tables, authored or reviewed drafts of the paper, and approved the final draft.

## Human Ethics

The following information was supplied relating to ethical approvals (i.e., approving body and any reference numbers):

Griffith University granted Ethical approval to carry out the study. (Ethical Application Ref: #2016/168).

## Data Availability

The raw data and that calculations that were not performed in SPSS are available in the Supplemental Files.

## Supplemental Information

Supplemental information for this article can be found online at http://dx.doi.org/10.7717/peerj.12192#supplemental-information.

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
