# Peer review of "Modernising tactile acuity assessment; clinimetrics of semi-automated tests and effects of age, sex and anthropometry on performance"

_PeerJ, doi:10.7717/peerj.12192_

## Round 0.1 · original submission · Major Revisions

Dear Authors, Your manuscript requires major revisions as per suggestions of the two peer reviewers.

·

Basic reporting

English is of a very high standard although I would check the opinion provided by the native English reviewer, too. Only some references are missing that could complete the whole context of the study. I pasted them in the box "general comments". Figures and Tables are accurate and provided raw data seems to be correct. I was able to reproduce reliability coefficients according to analytical descriptions while doing a 'cross-check' of analyses. Methodologically sound paper.

Experimental design

Maybe a clear statement in the introduction about the gap in the literature would be a suggestion for improvement. The study is definitely of high ethical standards. Analysis and data are available for review. Regarding the description of the method, I found it very clear, only some procedural aspects are missing which in my opinion can improve replication of experiments (see comment in general comments).

Validity of the findings

Data are provided for review. Do authors plan to make them publicly available? I assume that the policy of the journal is to make access to the data easy and in the spirit of the open science framework. Conclusions should be extended by results of internal consistency analysis.

Additional comments

Thank you very much for the opportunity to review the manuscript by Olthof et al. It is an extremely interesting study that places the reinvigorated field of tactile acuity in the context of neck region and neck-related pain. The manuscript presents two independent studies, one with state-of-the-art reliability analysis of 2-point discrimination threshold and an innovative novel testing procedure that eliminates examiner and caliper-related error. The second study is dedicated to what authors refer to as the "internal consistency" of the test. The second study also targets factors that can potentially confound the results (e.g. mentioned floor effects).

In general, I have to congratulate the authors for this methodologically sound series of technical studies. Several novel aspects can be found here. In experiment 1, the iTAD has been contrasted in terms of inter-rater reliability analysis showing slightly higher ICCs for iTAD procedure than 2PD. I commend the authors for providing extensive reports including SEMs and different CIs possibilities while reporting SDC. In experiment 2, the authors found high internal consistency (ICCs > 0.84) and consistent pattern in prediction models: age and sex explained variance similarly in two calculated iTAD scores.

I have a few primary concerns that I think the authors can easily address:

i. Can the authors explain the importance of moving only one caliper tip having the other one 'fixed' in terms of stimulation site? Could it be that tactile corpuscles adapt heavily in the fixed site, leading to a larger discrimination threshold (as the subject could not feel that point clearly)? Where the mean and SD comparable to data from previous studies testing sensibility of the neck region? I would report it in the discussion. In addition, I really like how the whole protocol for 2PD is presented in the figure. I found it very informative and easy to reproduce.

ii. It is not clear from the description how the training of the raters was performed. Did they listen to instructions? Did they assess themselves as a part of testing? Or maybe there was another pilot form of assessment before the study commenced? It would be beneficial if the authors could add some more information, as it could further expand the existing literature on how to prepare the clinician for the assessment.

iii. ICC - measurement of the 2PD is actually being represented by the average of 4 runs. Would it be preferable to perhaps change the model? Also, I am wondering whether CoV (Table 1 data) should be calculated using the traditional formula with SD placed in the nominator instead of SEM (Lovie, 2005). Could the authors explain their choice or change the presenting method?

iv. The authors might consider discussing two aspects: a) some attempts have been made to make tactile acuity assessment automatic (for example: Frahm et al. 2021, Yokota et al. 2020). Can the authors contrast their results with mentioned methods?

v. I think it would be interesting to simply correlate 2PD data with iTAD data (btw. I think Apple will be very happy if the app for the device is released one day). It can further strengthen the paper by providing data on the validity of the novel test.

vi. In 2015 I thought that 2PD was a real horror. Many procedural aspects affected the outcome, thus I tried to somehow find alternatives. In the end, 2PD was winning as it is relatively easy and has a long tradition of application (since work done by Ernst Weber). Today I'm still struggling with this test and always return to the study by Tong et al. (2013). Maybe all of us failed. Maybe 2PD is too simplistic to measure discrimination over complex irregularities of receptive fields (Johansson & Vallbo, 1983). Could the grating-orientation-task be the abandoned solution? I would ask the authors to comment whether the iTAD has potential to measure acuity with a higher degree of internal validity compared to simple 2PD.

Small comments and suggestions

1. Introduction: "Patients with persistent pain typically demonstrate poorer tactile acuity than healthy controls, which is thought to reflect changes in somatosensory processing related to the condition" - authors might consider citing meta-analysis by Mark Catley (2014).

2. Were the locations for the measurements determined by each examiner separately? I would add this information.

3. The sample size description is very honest and transparently reported.

4. Lines 117 and 130: I would make consistent explanations as to why the acuity is better OR higher. Sometimes explanations refer to instances of "better acuity" and sometimes "lower acuity".

5. Was the assessor trained in study 2 as well?

6. "...values were somewhat better" - I would say " higher".

7. Tricky question: Do the authors think that previous intra-rater reliability would change with the updated prototype?

8. Mediations: Which was applied - Baron & Kenny method or Bootstrapping? (If so, the sampling should be reported).

References

1. Lovie, P. Encyclopedia of Statistics in Behavioral Science. (2005) doi:10.1002/0470013192.bsa107.

2. Yokota H, Otsuru N, Kikuchi R, Suzuki R, Kojima S, Saito K, Miyaguchi S, Inukai Y, Onishi H. Establishment of optimal two-point discrimination test method and consideration of reproducibility. Neurosci Lett. 2020 Jan 1;714:134525.

3. Frahm KS, Gervasio S. The two-point discrimination threshold depends both on the stimulation noxiousness and modality. Exp Brain Res. 2021 May;239(5):1439-1449.

4. Johansson, R. S. & Vallbo, Å. B. Tactile sensory coding in the glabrous skin of the human hand. Trends Neurosci 6, 27–32 (1983).

Reviewer 2 ·

Basic reporting

see comments to authors below:

Relevance of measuring tactile acuity at the neck is not made clear, Figure 3 appears redundant while figures showing the distribution of the individual scores is missing, raw data should be provided in a generic data format

Experimental design

see comments to authors below:

some methods details are missing

Validity of the findings

no comment

Additional comments

The study aimed at assessing clinimetric properties of the iTAD device for measuring tactile acuity with vibrotactile stimuli at the neck. Exp. 1 compared inter-rater reliability and measurement error between iTAD and more traditional two-point discrimination scores. Exp. 2 tested internal consistency and floor/ceiling effects of iTAD scores in a larger sample of n=100. In addition, relationships of iTAD scores with age, gender, BMI and neck surface area were tested. It was found that inter-rater reliability was moderate-to-good for iTAD and measurement error was lower than for TPD. Exp. 2 results showed high internal consistency, no overt floor/ceiling effects as well as a direct relation of iTAD scores with age and gender which was not mediated by BMI or neck surface area. Results are interpreted as showing suitability of iTAD device for assessing tactile acuity in clinical settings, in particular for assessing change in tactile acuity due to interventions.

Overall, I think methods-oriented studies like this are important to assess reliability and validity of psychophysical tests, but are somewhat lacking for tactile acuity measures. Thus, the present study is a welcome addition to the literature. However, there are several issues with the present manuscript that would need to be addressed in a revision:

Introduction:

The rationale why measuring tactile acuity at the neck is important or relevant (in a clinical context) is not clear. There are excellent alternatives to the traditional TPDT for measuring tactile acuity at the fingertip which outperform TPDT in terms of reliability and validity, e.g. the commercially available JVP domes introduced by Van Boven and Johnsen or tactile acuity charts (Legge et al., 2008, Percept Psychophys; Bruns et al., 2014, PLoS ONE). However, these measures are not discussed at all in the paper.

It might also be helpful to introduce the iTAD procedure (and how it differs from traditional TPDT) in more detail already in the introduction, I think that many readers will not be familiar with this method.

Methods:

There are a number of important information on the procedure missing that would be required if one were to replicate the study:

TPDT: What is the size/diameter of the two needles (the part that actually contacts the skin)? What was the approximate indentation depth and duration of stimulation?

It seems that one needle (as a control condition) was never applied, i.e. there were always two needles present and the correct answer would have been “two” in every trial?

Needle distance for ascending run is given (15mm, increasing with 5mm steps), but what about the descending runs?

What does "Before running the reversed direction, the distance was overshot by 10mm" mean? Does this mean that the start distance of subsequent run was determined by the reversal point of the previous run plus/minus 10 mm?

iTAD: no information regarding vibration frequency, strength and duration is provided

Trial procedure is not entirely clear, e.g. could only neighboring locations be activated in the direction task?

Section 2.1.2. participants Exp. 1: all essential information (number of participants, age and gender distribution, handedness) is entirely missing. Sample of Exp. 2 could also be described in more detail.

Figure 3 might be redundant because mediation analysis model does already become clear from Figure 4.

Results:

Unfortunately, any graphical presentation of the (individual) results is missing. For Exp. 2, it would be desirable to see the distribution of individual performance as a function of age (with gender color-coded).

For Exp. 1, an analysis of the relationship between TPDT and iTAD is missing. Arguably e.g. localization of one point on skin (as in iTAD localization task) measures a different property than discriminating one from two points (see e.g. Dinse et al., 2008, in Grunwald (Ed.), Human haptic perception). Thus, it would be important to establish the agreement between TPDT and iTAD measurements (i.e., do participants with high acuity in TPDT also show high acuity in iTAD?), in particular because the authors suggest the iTAD as a replacement of TPDT in clinical settings. Graphically, the agreement of the two methods could e.g. be shown using the limits of agreement approach introduced by Bland and Altman (see Bruns et al., 2014, PLoS ONE, for an example with different tactile acuity measures).

Discussion:

iTAD (unlike TPDT) was administered with an automated computer-controlled procedure. Thus, wouldn't assessor be expected to play less of a role than in TPDT? Yet ICC values are fairly similar between the two methods, I wonder whether this result actually indicates lower test-retest reliability (independent of assessor) than for TPDT?

Supplementary material:

Raw data is provided in SPSS format. A generic data format (e.g., txt or csv) would be preferable.

---

## Round 0.2 · accepted · Accept

Thank you for your revised manuscript which has been accepted. Congratulations!

Reviewer 2 ·

Basic reporting

no comment

Experimental design

no comment

Validity of the findings

no comment

Additional comments

The authors have done a great job in further improving their manuscript. All comments from the previous round of reviews have been addressed.